# The Adenylate Cyclase (CyaA) Toxin from *Bordetella pertussis* Has No Detectable Phospholipase A (PLA) Activity In Vitro

**DOI:** 10.3390/toxins11020111

**Published:** 2019-02-13

**Authors:** Alexis Voegele, Mirko Sadi, Dorothée Raoux-Barbot, Thibaut Douché, Mariette Matondo, Daniel Ladant, Alexandre Chenal

**Affiliations:** 1Chemistry and Structural Biology Department, Institut Pasteur, UMR CNRS 3528, CEDEX 15, 75724 Paris, France; alexis.voegele@pasteur.fr (A.V.); mirko.sadi@outlook.de (M.S.); dorothee.raoux-barbot@pasteur.fr (D.R.-B.); 2Université Paris Diderot Paris VII, 75013 Paris, France; 3Mass Spectrometry for Biology Unit, Proteomics Platform, Institut Pasteur, USR CNRS 2000, CEDEX 15, 75724 Paris, France; thibaut.douche@pasteur.fr (T.D.); mariette.matondo@pasteur.fr (M.M.)

**Keywords:** adenylate cyclase toxin, CyaA toxin, bordetella pertussis, phospholipase A, fluorescence

## Abstract

The adenylate cyclase (CyaA) toxin produced in *Bordetella pertussis* is the causative agent of whooping cough. CyaA exhibits the remarkable capacity to translocate its N-terminal adenyl cyclase domain (ACD) directly across the plasma membrane into the cytosol of eukaryotic cells. Once translocated, calmodulin binds and activates ACD, leading to a burst of cAMP that intoxicates the target cell. Previously, Gonzalez-Bullon et al. reported that CyaA exhibits a phospholipase A activity that could destabilize the membrane to facilitate ACD membrane translocation. However, Bumba and collaborators lately reported that they could not replicate these results. To clarify this controversy, we assayed the putative PLA activity of two CyaA samples purified in two different laboratories by using two distinct fluorescent probes reporting either PLA2 or both PLA1 and PLA2 activities, as well as in various experimental conditions (i.e., neutral or negatively charged membranes in different buffers.) However, we could not detect any PLA activity in these CyaA batches. Thus, our data independently confirm that CyaA does not possess any PLA activity.

## 1. Introduction

The adenylate cyclase (CyaA or ACT) toxin is secreted by *Bordetella pertussis*, the causative agent of whooping cough. The CyaA toxin is one of the major virulence factors produced by *B. pertussis* and is involved in the early stages of respiratory tract colonization [1,2,3,4,5]. CyaA invades eukaryotic target cells, where it is activated by calmodulin (CaM) to produce supra-physiological cAMP concentrations. It is a 1706-residue long protein that belongs to the Repeat-in-ToXin (RTX) family of cytolysins [6,7,8,9,10]. CyaA is a multi-domain protein [11,12], harboring from the N- to C-terminus, that has (i) a calmodulin-activated catalytic domain, ACD, located in the first 364 N-terminal residues [13,14,15]; (ii) a translocation region (TR, residues 365 to 527) that is essential for ACD translocation into target cells [16,17,18,19]; (iii) a hydrophobic region (HR, residues 528 to 710) that inserts into the cell membrane and makes cation-selective pores leading to cell lysis [10,20,21]; (iv) an acylation region (AR, residues 711 to 1005) that contains two post-translational acylation sites on K860 and K983 [22,23,24] (modifications are required for the proper refolding and activity of CyaA) [5,12,22,23,25]; and (v) a C-terminal domain (RD, residues 1006 to 1706) made of about 40 copies of calcium-binding RTX motifs [8,9,10,26] that are required for RD and subsequently CyaA folding [27,28,29,30,31,32,33,34,35] and interaction with the CyaA cell receptor, CD11b/CD18 [36,37]. CyaA uses an original intoxication mechanism: once secreted by *B. pertussis*, it invades eukaryotic cells through a unique molecular mechanism that involves a direct translocation of its ACD catalytic domain across the plasma membrane. Yet, the molecular mechanisms of this unique entry process remain largely unknown.

Recently, H. Ostolaza et al. reported that CyaA exhibits a phospholipase A (PLA) activity [38,39] that is involved in destabilizing the lipid bilayer to facilitate ACD translocation across the plasma membrane. These results were contested by Bumba et al., who subsequently reported that CyaA has no PLA1 activity and showed that the two residues Ser606 and Asp1079 claimed to be critical for the putative PLA activity of CyaA, were not involved in CyaA translocation [40]. 

To clarify this controversy, we examined the PLA activity of two purified CyaA samples, one from our laboratory (CyaA_IP_) and the second provided by Ostolaza’s team (CyaA_UBC_). Mass spectrometry data revealed that both CyaA batches were devoid of contaminating *E. coli* phospholipase. Using two distinct fluorescent probes of PLA activity, PED6 (specific to PLA2 activity) and B3781 (for both PLA1 and PLA2 activities), we could not detect any PLA activity in these CyaA preparations. This unambiguously indicates that CyaA has no detectable phospholipase A activity in vitro, which agrees with the prior results of Bumba et al. [40]. We discuss potential artifacts that might have blurred the conclusions of the Ostolaza team. 

## 2. Results

### 2.1. Characterization of the CyaA_IP_ and CyaA_UBC_ Preparations

The CyaA preparations used in this study, CyaA_IP_ from our laboratory and CyaA_UBC_ from Ostolaza’s team, were recombinantly produced in *E. coli* and purified using previously described protocols [12,38,41]. Both protein samples showed potent hemolytic activity and cell invasive capacities when tested on sheep erythrocytes (data not shown). Proteomic analysis of the CyaA batches was then performed to determine the extent of contamination of the samples by *E. coli* host phospholipases. The total *E. coli* contaminant proteins from both CyaA batches were identified using liquid chromatography–mass spectrometry (LC-MS/MS) (see Appendix A). The *E. coli* phospholipases (protein references EC 3.1.1.32 and EC 3.1.1.4) were not detected among contaminants in both CyaA samples. A lysophospholipase (EC 3.1.1.5), however, was identified in CyaA_UBC_ but at very low abundance (see Appendix A). Hence, we can conclude that both CyaA_IP_ and CyaA_UBC_ are devoid of any contaminating PLA from *E. coli*. 

### 2.2. PLA Activity Measurements as Monitored with the PED6 Fluorescent Probe

We first investigated PLA activity using a fluorescent probe, PED6, which is specific for PLA2 activity. In PED6, a 4,4-Difluoro-5,7-Dimethyl-4-Bora-3a,4a-Diaza-s-Indacene group linked to the acyl chain is quenched with a 2,4-Dinitrophenyl group linked to the lipid headgroup. The cleavage of PED6 using a PLA2 enzyme is monitored by the fluorescent recovery of the indacene group (excitation at 480 nm; emission at 490–650 nm; full spectra are shown in Appendix A). Figure 1A shows a time course recording of the fluorescence emission at 515 nm (F_515_) of PED6 in the presence of LUV (DOPC:PED6 at a 8:2 molar ratio) in buffer alone or after addition of various proteins. Further, the addition of 10 nM crotoxin, a neurotoxin from *Crotalus durissus terrificus* [42,43] with a potent PLA activity, induced a large increase (up to 15 fold) in fluorescence emission (Figure 1A,B, black trace). In contrast, no detectable PED6 fluorescence changes were observed after the addition of 10 nM BSA, a control protein lacking PLA activity, or, more importantly, of 10 nM of either CyaA_IP_ or CyaA_UBC_ (Figure 1A, orange and red traces, respectively). This indicates that the two CyaA proteins do not exhibit PLA2 activity. Figure 1B shows the same kinetics of fluorescence emission after normalization to the fluorescence intensity at a time of zero. While a high increase in fluorescence is observed with crotoxin (black trace), the fluorescence intensity variations observed with the two CyaA samples over a 30 min course are within a few percent of the initial F_515_ (Figure 1B, inset) and are essentially indistinguishable from that recorded with the buffer alone, or upon addition of BSA or of 60 mM urea (corresponding to the residual urea concentration obtained after addition of 10 nM CyaA proteins, as the toxins were stored in concentrated urea solutions). These data unambiguously indicate that the CyaA samples had no detectable PLA2 activity in these conditions. 

Figure 1C,D show a different representation of the same data, plotted as relative ratio of fluorescence intensity at 515 nm over fluorescence intensity at 575 nm (F_515_/F_575_) for all samples. These plots (before, Figure 1C, and after, Figure 1D, normalization of baseline to 1 at time t = 0, respectively) fully confirm that only the crotoxin sample displays a significant PLA2 activity, while with other protein samples, CyaA_IP_, CyaA_UBC_, or BSA, the basal, random drifts of F_515_/F_575_ fluorescence ratio are indistinguishable from that recorded with the buffer alone or 60 mM urea. 

### 2.3. PLA Activity Measurements as Monitored with the B3781 Fluorescent Probe

We then investigated the PLA activity using the B3781 fluorescent probe that reports both PLA1 and PLA2 phospholipase activities (Figure 2). Fluorescence excitation was fixed at 342 nm and the emission spectra of the B3781 probe were recorded from 350 to 600 nm each minute for 30 min (see Appendix A). The 1-pyrenebutanoyl groups linked to the lipid acyl chains form excimers was characterized using a maximum fluorescence emission near 480 nm. Hydrolysis of lipid acyl chains using a phospholipase releases a pyrene fatty acid that diffuses away from the pyrene lysophospholipid. These two pyrene derivatives exhibit a maximum emission of fluorescence between 380 and 400 nm [44,45].

Figure 2 shows a time course recording of the fluorescence emission changes (at 380 nm, F_380_, Panel A, or 479 nm, F_479_, Panel B) of B3781 (in LUV composed of POPC:POPG:B3781 at a lipid ratio 79:20:1) after the addition of crotoxin, CyaA_IP_, or urea (180 mM final concentration; full spectra are shown in Appendix A). Large changes in fluorescence, typical of excimer-to-monomer conversion of the B3781 probe, were observed with crotoxin (100 nM) as a result of its PLA activity. However, no detectable changes were noticed with up to 600 nM of purified CyaA, demonstrating the total lack of phospholipase activity of the toxin. Panels C and D in Figure 2 show the same results, as a ratio of fluorescence intensity at 380 nm over fluorescence intensity at 479 nm, F_380_/F_479_, before and after normalization to the initial F_380_/F_479_ value at a time of zero. All data representations clearly indicate that the CyaA toxin does not exhibit any detectable phospholipase A activity in these in vitro conditions.

## 3. Discussion

It is clear from our present data that no significant PLA1 or PLA2 activity can be detected for highly purified CyaA proteins using fluorescent probes classically used for measuring phospholipase A activity. These experiments were repeated several times, using several distinct CyaA batches, as well as different batches of LUV of different composition—in the absence (POPC:PED6 at 8:2 molar ratio) or in the presence of anionic lipids (POPC:POPG: B3781 at 79:20:1 molar ratio)—and different buffer conditions (Tris-HCl instead of HEPES-Na, data not shown). Our PLA assays were validated using crotoxin, a venom neurotoxin with potent PLA activity, and BSA as positive and negative controls, respectively. We therefore concluded that CyaA has no detectable PLA activity in vitro. These data corroborate the prior results found by Bumba et al. [40], who likewise could not detect PLA activity in their purified batches of CyaA.

Yet, how can these results be reconciled with the claims of the Ostolaza study? These authors used a similar fluorescent probe, PED1 to monitor the PLA1 activity of CyaA. In their reported experiments, only very small changes in fluorescent intensity of the probe (<1.1–1.4 relative ratio) were observed after the addition of CyaA. Moreover, the kinetic experiments presented in their recent paper [39] revealed odd profiles that question the validity of interpretation of the fluorescence changes as signatures of a PLA enzymatic activity. Indeed, they observed an abrupt, yet limited, change of fluorescence intensity just after addition of the CyaA sample (i.e., at the first recording point) with no further changes during the following 30 min of incubation. Strikingly, all assays at various concentrations of CyaA showed an identical profile: an abrupt initial change followed by a constant plateau over the whole course of the kinetics. Only the plateau values were different and roughly proportional to the amount of added proteins. Surprisingly, the authors were able to derive “velocities” (i.e., fluorescent change per minute) from these odd kinetic data, as shown in their plot of Figure 2C in [39]. 

The authors did notice the singular feature of the CyaA PLA activity. To explain this, as well as the very weak lipid hydrolytic activity as compared to that of other known phospholipases A, they proposed that a local accumulation of the lysophospholipidic product around the toxin molecules in the membrane could inhibit its PLA catalytic activity. Yet, it is hard to understand why the lysophospholipid could not continuously diffuse out in the membranes while being replaced by new substrates, i.e., unprocessed phospholipidic PLA probes. Furthermore, this hypothesis would require that the CyaA toxin binds to the lipid vesicles and hydrolyzes the PLA fluorescent probes within seconds after being added to the membrane with no further interaction and/or hydrolytic events occurring during the further 30 min of incubation. This is obviously not the case as all prior kinetic measurements of CyaA activities (including in prior publications from Ostolza and collaborators) clearly showed a progressive time-dependent accumulation of cAMP, hemolysis, or membrane permeabilization over a 5–30 min time scale. Therefore, it is more reasonable to assume that the weak and fast fluorescence increase observed by Gonzalez-Bullon et al. was an artifact, due to photon counting variations induced by light scattering as a result of the CyaA sample addition to the solution rather than to the consequence of a putative enzymatic cleavage of the probe by the PLA activity of CyaA. Indeed, in our experiments we did not observe any similar small burst of fluorescence after adding CyaA, but only a very limited random drift that was not different from that observed with the negative control BSA protein. In summary, on the basis of our in vitro assays with fluorescent PLA probes, we can unambiguously conclude that CyaA has no detectable PLA activity in vitro, which agrees with the prior studies of Bumba et al. [40].

## 4. Material and Methods

### 4.1. Buffers

Buffer A is composed of 20 mM HEPES, 150 mM NaCl and 10 mM CaCl_2_ pH 7.4. Buffer B is composed of 20 mM HEPES, 150 mM NaCl, and 2 mM CaCl_2_ pH 7.4.

### 4.2. Protein Samples 

CyaA is expressed in *E. coli*, purified to homogeneity by established procedures [12,41]. CyaA was stored in 20 mM HEPES, 6 M urea, pH 7.4. CyaA was purified in our laboratory at Institut Pasteur and is hereafter called CyaA_IP_. The second batch of CyaA used in this study was a kind gift from Helena Ostolaza, from the University of Basque Country, hereafter named CyaA_UBC_. The crotoxin protein from *Crotalus durissus terrificus* is a kind gift from G. Faure [42,43]. Crotoxin was resuspended in 0.1% BSA in buffer B. BSA (from Sigma, St Louis, MI, USA, reference A0281) was resuspended in buffer B.

### 4.3. Mass Spectrometry Analysis

#### Sample Preparation

The protein digestion of the CyaA_IP_ batch was performed as follows: CyaA was placed in 20 mM HEPES, 6 M in urea, and the pH 7.4 was digested by Lys-C (V1671, Promega, Madison, WI, USA) at a Lys-C/CyaA ratio 1/50 (w/w), which corresponded to a 1/7 molar ratio (mol/mol). The sample was vortexed and centrifuged for 1 min at 3000 rpm then incubated for 3 h at 37 °C under agitation (500 rpm). Urea concentration decreased to 2 M with the addition of 20 mM HEPES, which optimized trypsin digestion. Trypsin (V5111, Promega, Madison, WI, USA) was added at a trypsin/CyaA ratio of 1/50 (w/w). The sample was vortexed and centrifuged for 1 min at 3000 rpm then the digestion was carried out for 1 h at 37 °C under agitation (500 rpm). The digestion stopped with 2% formic acid (FA) (Fluka, 94318) and digested peptides were desalted on reversed phase C_18_ Sep-Pak Cartridge (WAT054955, Waters Corporation, Milford, MA, USA). Peptides were eluted with a final 80% Acetonitrile (ACN)/0.1% Formic acid (FA). Resulting peptides were vacuum centrifuged to evaporate ACN and then lyophilized. Peptide samples were reconstituted in 2% ACN/0.1% FA.

Protein digestion of the CyaA_UBC_ batch was performed as follows: 15 µg of the purified fraction containing CyaA_UBC_ were diluted with Urea 8 M/NH_4_HCO_3_ 100 mM. The samples were reduced with 5 mM TCEP over 1 h and alkylated with 20 mM iodoacetamide over 1 h. The sample was digested with rLys-C (Promega, Madison, WI, USA, V1671), ratio 1:50 (enzyme:protein/w:w) for 4 h at 37 °C and then with Sequencing Grade Modified Trypsin (Promega, Madison, WI, USA, V5111), ratio 1:50 (enzyme:protein/w:w) overnight at 37 °C. The digestion stopped with 2% formic acid (FA) (Fluka, 94318) and digested peptides were purified with C_18_ Spin Columns Pierce™ (ThermoFisher Scientific, Waltham, MA, USA, 89870). Peptides were eluted with 80% ACN/0.1% FA. Resulting peptides were vacuum centrifuged to evaporate ACN and then lyophilized. Peptide samples were reconstituted in 2% ACN/0.1% FA.

### 4.4. LC-MS/MS Analysis of the CyaA Batches

LC-MS/MS analysis of the CyaA_IP_ batch was performed on a U3000-Velos mass spectrometer. A nanochromatographic system Ultimate 3000 system (Dionex, Amsterdam, The Netherlands) was coupled on-line to a LTQ-Orbitrap Velos Mass Spectrometer (Thermo Fisher Scientific, Waltham, MA, USA) using an integrated column oven (PRSO-V1, Sonation GmbH, Biberach, Germany). For each sample, 1.0 μg of peptides were loaded onto a C_18_ pre-column (300 μm inner diameter × 5 mm; Dionex, Sunnyvale, CA, USA) at 30 μL/min in 2% ACN, 0.1% FA. After 5 min of desalting, the pre-column was switched online to an in-house packed 15 cm nano-HPLC column (75 μm inner diameter) with C_18_ resin (3 μm particles, 100 Å pore size, ReproSil-Pur Basic C18, Dr. Maisch GmbH, Ammerbuch-Entringen, Germany). Peptides were then separated with a multi-step gradient from 2% to 9% buffer B (80% ACN/0.1% FA) in 5 min, 9% to 29% buffer B in 70 min, 29% to 56% buffer B in 30 min, and 56% to 100% buffer B in 5 min at a flow rate of 300 nL/min over 132 min. The column temperature was set to 40 °C. MS data were acquired using Xcalibur software with a survey scan (300–2000 m/z) analyzed into the Orbitrap mass analyzer at a resolution setting of 60,000, followed by 10 CID fragmentations analyzed into the linear ion trap. The AGC targets for MS and MS/MS scans were set to 1E6 and 5E3 respectively. The isolation width was set to 2.5 m/z and the normalized collision energy was set to 35. Selected ions were dynamically excluded for 30 s.

LC-MS/MS analysis of the CyaA_UBC_ batch was performed on a Proxeon 1200-Q Exactive mass spectrometer. A nanochromatographic system (Proxeon EASY-nLC 1200, Thermo Fisher Scientific, Waltham, MA, USA) was coupled on-line to a Q ExactiveTM Plus Mass Spectrometer (Thermo Fisher Scientific, Waltham, MA, USA) using an integrated column oven (PRSO-V1, Sonation GmbH, Biberach, Germany). For each samples, 1.0 μg of peptides was injected onto a 46 cm homemade C_18_ column (1.9 μm particles, 100 Å pore size, ReproSil-Pur Basic C18, Dr. Maisch GmbH, Ammerbuch-Entringen, Germany) and separated with a multi-step gradient from 2% to 7% buffer B (80% ACN/0.1% FA) in 5 min, 7% to 23% buffer B in 70 min, 23% to 45% buffer B in 30 min, and 45% to 95% buffer B in 5 min at a flow rate of 250 nL/min over 132 min. The column temperature was set to 60 °C. MS data were acquired using Xcalibur software and a data-dependent Top 10 method with survey scans (300–1700 m/z) at a resolution of 70,000 and a MS/MS scans (fixed first mass 100 m/z) at a resolution of 17,500. The AGC target and maximum injection time for the survey scans and the MS/MS scans were set to 3E6, 20 ms, and 1E6, 60 ms, respectively. The isolation window was set to 1.6 m/z and normalized collision energy fixed to 28 for HCD fragmentation. We used a minimum AGC target of 1.0E4 for an intensity threshold of 1.7E5. Unassigned precursor ion charge states as well as 1, 7, 8, and >8 charged states were rejecte and the peptide match was disabled. Excluded isotopes were enabled and selected ions were dynamically excluded for 45 s.

### 4.5. MaxQuant Analysis

Raw data were analyzed using MaxQuant software version 1.5.1.2 [46] using the Andromeda search engine [47]. Based on the protein expression system, the MS/MS spectra were searched against the *Escherichia coli* strain BL21DE3 UniProt database containing 4,156 proteins plus the CyaA sequence. The digestion mode was set to trypsin and a maximum of two missed cleavages were allowed. N-terminal acetylation, Methionine oxidation, and Lysine N-acylations (Myristoyl-4H, Myristoleylation, Myristoylation, Palmitoleylation, Palmitoylation) were set to variable modifications while Cystein Carbamidomethylation was set as fixed modification. The minimum peptide length was fixed to amino acids and the required false discovery rate was set to 1% at the PSM level and protein level. The main search peptide tolerance was set to 4.5 ppm and to 0.5 Da for the MS/MS match tolerance. Second peptide was enabled to identify co-fragmentation events.

### 4.6. Fluorescent Probes

PED6 (N-((6-(2,4-Dinitrophenyl)amino)hexanoyl)-2-(4,4-Difluoro-5,7-Dimethyl-4-Bora-3a, 4a-Diaza-s-Indacene-3-Pentanoyl)-1-Hexadecanoyl-sn-Glycero-3-Phosphoethanolamine, Triethylammonium Salt), and B3781 (1,2-Bis-(1-pyrenebutanoyl)-sn-glycero-3-phosphocholine) probes were purchased from Thermo Scientific. The PED6 and B3781 probes were resuspended in chloroform and stored at −20 °C under Argon.

### 4.7. Preparation of PED6-Containing LUV

Large unilamellar vesicles (LUV) were composed of DOPC:PED6 at a 8:2 molar ratio and prepared by reverse phase evaporation followed by extrusion of the multi-lamellar vesicles (MLV) suspension through 200 nm pore size polycarbonate filters [19]. A solution of 1 mL of LUVs was prepared at a lipid concentration of 500 µM in buffer A. Size, polydispersity, and electrophoretic mobility of LUV suspensions were measured on a Zetasizer NanoZS (Malvern, UK). All proteins (CyaA samples, BSA, and crotoxin) were assayed for PLA activity at a final protein concentration of 10 nM. Lipid concentration of LUV in the cuvette was 10 µM, i.e., a lipid:protein molar ratio of 1000. Experiments were performed under constant stirring at 37 °C using a 109 004F cuvette (Hellma) in a Jasco fluorometer. Fluorescence excitation was fixed at 480 nm and the emission spectra were recorded from 490 to 650 nm each minute for 30 min. The 2,4-Dinitrophenyl group linked on the lipid headgroup does quench the 4,4-Difluoro-5,7-Dimethyl-4-Bora-3a,4a-Diaza-s-Indacene group linked to the acyl chain. PLA2 activity will allow fluorescence recovery of the indacene derivative released in the membrane and diffusing away from the 2,4-Dinitrophenyl group liked to the lysophospholipid. 

### 4.8. Preparation of B3781-Containing LUV 

Large unilamellar vesicles (LUV) were composed of POPC:POPG:B3781 at a 79:20:1 molar ratio and prepared by reverse phase evaporation followed by extrusion of the multi-lamellar vesicles (MLV) suspension through 200 nm pore size polycarbonate filters [19]. A solution of 2 mL of LUVs was prepared at a lipid concentration of 2 mM in buffer B. Size, polydispersity, and electrophoretic mobility of LUV suspensions were measured on a Zetasizer NanoZS (Malvern). CyaA and crotoxin were assayed for PLA activity at a final protein concentration of 600 nM and 100 nM, respectively. Lipid concentration of LUV in the cuvette was 200 µM, i.e., a lipid:protein molar ratio of 333 for CyaA and 2000 for crotoxin. Experiments were performed under constant stirring at 37 °C using a 109 004F cuvette (Hellma) in a Jasco fluorometer. Fluorescence excitation was fixed at 342 nm and the emission spectra were recorded from 350 to 600 nm each minute for 30 min. The 1-pyrenebutanoyl groups linked to the lipid acyl chains form excimers characterized by a maximum emission of fluorescence near 480 nm. Hydrolysis of acyl chains by a phospholipase releases a pyrene fatty acid that diffuses away from the pyrene lysophospholipd. These two pyrene derivatives exhibit a maximum emission of fluorescence between 380 and 400 nm [44,45].

## Figures and Tables

**Figure 1 toxins-11-00111-f001:**
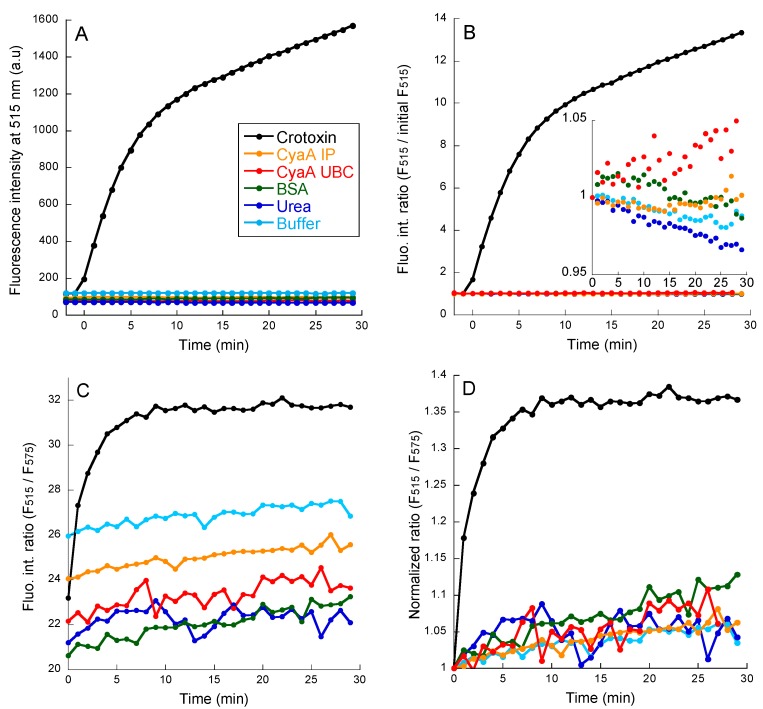
The kinetics of fluorescence changes of PED6 incorporated in DOPC:PED6 8:2 LUV (data from Appendix A). The time 0 min corresponds to the addition of the following samples: buffer (light blue), urea (dark blue), BSA (green), Crotoxin (black), CyaA_IP_ (orange), and CyaA_UBC_ (red). (**A**) Increase of fluorescence emission intensity at 515 nm. (**B**) Ratio of fluorescence intensity at 515 nm, F_515_, at time t normalized by initial fluorescence at 515 nm. Inset shows same data with an expanded *Y*-axis (from 0.95 and 1.05). (**C**) Ratio of fluorescence intensity at 515 nm over fluorescence intensity at 575 nm: F_515_/F_575_. (**D**) The F_515_/F_575_ values are normalized to the initial F_515_/F_575_ value at 0 min (see methods for details).

**Figure 2 toxins-11-00111-f002:**
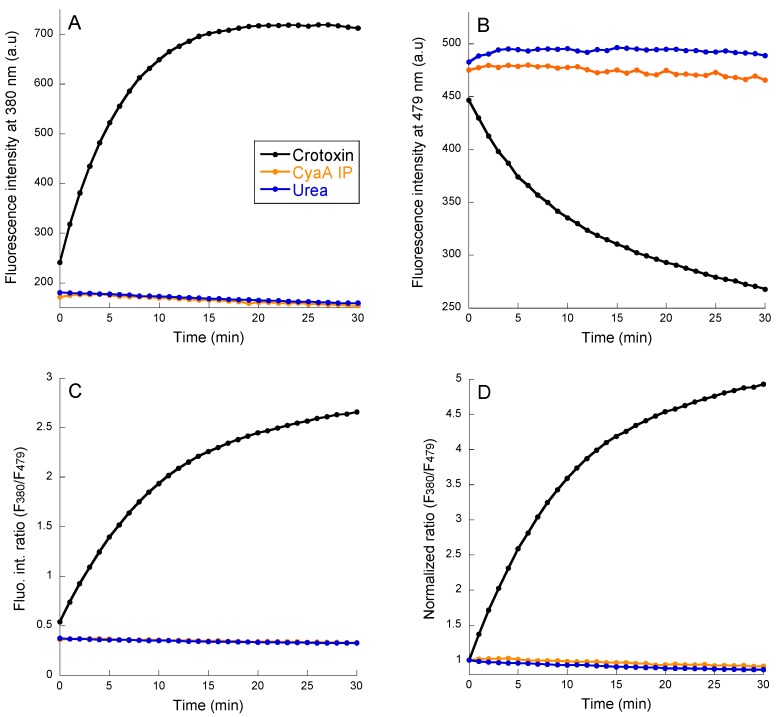
Kinetics of fluorescence changes of B3781 incorporated in POPC:POPG:B3781 79:20:1 LUV (data from Appendix A) upon addition of 100 nM crotoxin (black), 600 nM CyaA_IP_ (orange), and 180 mM urea (dark blue). (**A**) Increase of fluorescence intensity emitted at 380 nm. (**B**) Decrease of fluorescence emission intensity at 479 nm. (**C**) Ratio of fluorescence intensity at 380 nm over fluorescence intensity at 479 nm: F_380_/F_479_. (**D**) F_380_/F_479_ values are normalized to the initial F_380_/F_479_ value at 0 min.

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
