# Peer review of "The Adenylate Cyclase (CyaA) Toxin from *Bordetella pertussis* Has No Detectable Phospholipase A (PLA) Activity In Vitro"

_toxins, 2019, doi:10.3390/toxins11020111_

Round 1
Reviewer 1 Report
This manuscript demonstrated that the adenylate cyclase toxin (ACT) from Bordetella pertussis has no detectable phospholipase A (PLA) activity regardless of the preparation of recombinant ACT (rACT). The authors checked the contamination of other E. coli proteins with PLA activity on the two different preparation of rACT, and they found both rACT were devoid of any contaminating PLA. Further, they measured the PLA activity of both rACT by using the fluorescent probe PED6 for PLA2 activity and B3781 for PLA1 and PLA2 activities. As a results, both rACT did not exhibit any detectable PLA activity in these in vitro conditions. The results are clear and could finish the controversy about the PLA activity of ACT.
Reviewer 2 Report
This work represents a major contribution in our understanding of the mode of action of the adenylate cyclase cyaA from Bordetella pertussis and toxins in general. The authors present indisputable and carefully conducted biochemical data, including the fair analysis of the toxin purified by Ostolaza's group, showing that cyaA has no detectable phospholipase activity.
This article will serve as a quality standard in terms of enzymology approaches to characterize future PLA activities in bacterial toxins.
I fully agree with the elements brought into the discussion.
I recommend to publish this excellent and well-written article.